# MOF-Derived Ultrathin Cobalt Molybdenum Phosphide Nanosheets for Efficient Electrochemical Overall Water Splitting

**DOI:** 10.3390/nano12071098

**Published:** 2022-03-27

**Authors:** Xiang Wang, Linlin Yang, Congcong Xing, Xu Han, Ruifeng Du, Ren He, Pablo Guardia, Jordi Arbiol, Andreu Cabot

**Affiliations:** 1Catalonia Institute for Energy Research (IREC), Sant Adrià de Besòs, 08930 Barcelona, Spain; wxiang@irec.cat (X.W.); lyang@irec.cat (L.Y.); congcongxing@irec.cat (C.X.); ruifengdu@irec.cat (R.D.); renhe@irec.cat (R.H.); pguardia@irec.cat (P.G.); 2Departament d’Enginyeria Electrònica i Biomèdica, Universitat de Barcelona, 08028 Barcelona, Catalonia, Spain; 3Catalan Institute of Nanoscience and Nanotechnology (ICN2), CSIC and BIST, Campus UAB, Bellaterra, 08193 Barcelona, Catalonia, Spain; xu.han@icn2.cat (X.H.); arbiol@icrea.cat (J.A.); 4ICREA, Pg. Lluis Companys, 08010 Barcelona, Catalonia, Spain

**Keywords:** MOF, phosphide, nanosheet, water splitting

## Abstract

The development of high-performance and cost-effective earth-abundant transition metal-based electrocatalysts is of major interest for several key energy technologies, including water splitting. Herein, we report the synthesis of ultrathin CoMoP nanosheets through a simple ion etching and phosphorization method. The obtained catalyst exhibits outstanding electrocatalytic activity and stability towards oxygen and hydrogen evolution reactions (OER and HER), with overpotentials down to 273 and 89 mV at 10 mA cm^−2^, respectively. The produced CoMoP nanosheets are also characterized by very small Tafel slopes, 54.9 and 69.7 mV dec^−1^ for OER and HER, respectively. When used as both cathode and anode electrocatalyst in the overall water splitting reaction, CoMoP-based cells require just 1.56 V to reach 10 mA cm^−2^ in alkaline media. This outstanding performance is attributed to the proper composition, weak crystallinity and two-dimensional nanosheet structure of the electrocatalyst.

## 1. Introduction

Hydrogen, with a high gravimetric energy density (142 MJ kg^−1^) and zero-carbon emissions, is both a key component in the chemical industry and a very appealing energy carrier for clean and sustainable energy storage and supply [1,2]. Since molecular hydrogen is not freely available in nature, it needs to be extracted from hydrogen-containing compounds. Currently, fossil fuels are the main source of H_2_, which involves the release of large amounts of carbon. The electrochemical water splitting is the main green alternative to produce H_2_, but it is seriously hampered by the high cost and insufficient durability of current electrocatalysts, based on scarce novel metals such as Pt, Ir and Ru [3,4,5,6], and the slow kinetics of the hydrogen and oxygen evolution reactions (HER, OER), which makes water electrolysis not competitive with steam reforming of natural gas or coal gasification processes [7,8,9].

In order to overcome the current challenges and enable the massive production of H_2_ by water splitting, substantial efforts were devoted to the development of high activity, stable and cost-effective electrocatalysts for overall water splitting (OWS). Among the wide range of materials proposed for alkaline water electrolysis, including metal oxides/hydroxides [10,11,12], chalcogenides [13,14,15,16], nitrides [17,18] and carbides [19,20], metal phosphides demonstrated particularly attractive catalytic performances [21,22]. The outstanding performance of phosphides was related to their high electrical conductivity, favorable electronic structure and high stability against corrosion [23,24]. Amongst phosphides, cobalt phosphide (CoP) exhibits an exceptional HER activity associated with a proper electronic structure and specifically to the effective capture of protons by the negatively charged phosphorous atoms [25,26]. Besides, CoP is also characterized by a very high OER activity associated with the high ability of the positively charged cobalt cations to adsorb oxygen intermediates while the negatively-charged P facilitates the desorption of O_2_ molecules [27,28]. However, the OWS in CoP still requires too large overpotentials for practical applications, mainly ascribed to the high dissociation energy of water and the sluggish OER kinetics involving a multi-electron transfer process [29,30].

An effective strategy to optimize a material’s performance is the introduction of an additional element that provides additional degrees of freedom to modulate its electronic structure and surface properties. Within transition metal phosphide electrocatalysts, additional metals enable a fine-tuning of the d-band position, optimizing the adsorption free energy of the reactants/intermediates/products and thus improving the catalytic activity and even stability. In this direction, Xiao et al. reported the HER catalytic activity of CoP to be boosted by vanadium doping [31]. The introduced V strongly interacts with the hosted Co atoms, enhancing the VCoP electron density and thus accelerating the HER. Other elements, such as Zn [32], Mn [33], Ni [34], Ce [35], Cr [36] and W [21], were also demonstrated to promote either the HER or OER through enhancing electron interactions [28].

Beyond composition, the structure, morphology and organization of the catalyst particles are key parameters defining the density, accessibility and activity of the catalytic sites, which ultimately determine the catalytic activity. In this regard, metal–organic frameworks (MOFs), with a crystalline and porous structure formed by metal ion/cluster bridged by organic ligands, were demonstrated to be excellent sacrificial templates to produce porous carbon-based nanomaterials with tuned composition and morphology [37,38,39,40].

Inspired by the above considerations, we rationally designed a novel and highly effective bifunctional electrocatalyst for OWS. This new catalyst is based on CoP, structured as 2D ultrathin nanosheets and derived from the ZIF-67 MOF. It includes Mo^6+^ as a high valence 4d transition metal ion, which ionic radius of 0.62 Å matches well with that of Co^3+^ (0.63 Å), thus allowing the substitution of Co^3+^ by Mo^6+^ within the CoP lattice [41,42]. We demonstrated here that the proposed porous nanosheet-based structure and the incorporation of Mo within the CoP lattice enable rapid water dissociation and effective and stable HER and OER performances.

## 2. Materials and Methods

### 2.1. Chemicals

Ammonium molybdate tetrahydrate ((NH_4_)_6_Mo_7_O_24_·4H_2_O, 90%), cobalt nitrate hexahydrate (Co(NO_3_)_2_·6H_2_O, 99.9%), potassium hydroxide (KOH, 85%), iridium(IV) oxide (IrO_2_, 99.9% metal basis) and Nafion (5 wt% within a blend of low aliphatic alcohols and water) were obtained from Sigma-Aldrich (St. Louis, MO, USA). 2-Methylimidazole (C_4_H_6_N_2_, 99%) was purchased from Acros Organics (Antwerp, Belgium). Analytical grade methanol, ethanol and isopropanol were obtained from different sources. Milli-Q water was produced using an Elga Purelab flex. All chemicals were used as received.

### 2.2. Preparation of ZIF-67

ZIF-67 was produced following a previously reported procedure with some modifications [16,43]. Briefly, 0.87 g Co(NO_3_)_2_·6H_2_O was dissolved in 30 mL of methanol to obtain a clear solution. Subsequently, the above solution was poured into 30 mL of methanol containing 1.97 g of 2-methylimidazole under vigorous stirring. After mixing completely, the solution was incubated for 24 h at room temperature. Purple precipitates were collected by centrifugation; they were washed with methanol three times and then dried at 60 °C overnight.

### 2.3. Preparation of Mo–Co MOFs

One hundred and twenty milligrams of as-prepared ZIF-67 powder was ultrasonically re-dispersed in 20 mL of ethanol. This solution was poured into 100 mL of an aqueous solution containing 50 mg, 100 mg and 200 mg of ammonium molybdate under continuous magnetic stirring. The mixture was then stirred vigorously for 24 h at room temperature. Lavender precipitates were collected by centrifugation, washed with water at least three times and freeze-dried overnight.

### 2.4. Preparation of CoP and CoMoP

The obtained ZIF-67 and Mo–Co MOFs powders were placed in a porcelain boat within a horizontal tube furnace. In another boat, a 20× mass amount of NaH_2_PO_2_·H_2_O was placed at the upstream side of the tube furnace. The material was then annealed at 350 °C under N_2_ flow. After calcination for 2 h, the final black products were denoted as CoP and CoMoP, respectively.

### 2.5. Structural Characterization

Powder X-ray diffraction (XRD) was performed on a Bruker AXS D8 Advance X-ray diffractometer (Bruker, Billerica, MA, USA) with Cu-Kα radiation (λ = 1.5406 Å). Scanning electron microscopy (SEM) analysis was conducted with a Zeiss Auriga microscope (Carl Zeiss, Jena, Germany) equipped with an energy dispersive spectroscope analyses (EDS) detector operating at 20 kV. Transmission electron microscopy (TEM), High-resolution TEM (HRTEM), Annular dark-field scanning transmission electron microscope (HAADF-STEM) and electron energy loss spectroscopy (EELS) analysis were obtained using a field emission gun FEI Tecnai F20 microscope (FEI, Hillsboro, OR, USA) with a Gatan Quantum filter (Pleasanton, CA, USA) at 200 kV. X-ray photoelectron spectroscopy (XPS) measurements were conducted on a SPECS using an Al anode XR50 source at 150 W and a 150 MCD-9 detector from Phoibos (SPECS, Berlin, Germany).

### 2.6. Electrochemical Measurements

Electrochemical characterization was performed in a standard three-electrode system using an electrochemical workstation (CHI 760E, CH Instruments, Shanghai, China) in 1 M KOH solution (PH = 14). A graphite rod counter electrode and a Hg/HgO reference electrode were employed. Electrochemical impedance spectroscopy (EIS) was measured within a frequency from 0.01 Hz to 10 kHz at 10 mV amplitude. The initial voltage was fixed at the overpotential required to obtain a current density of 10 mA cm^−2^. The electrochemically active surface area (ECSAs) was determined using the electrochemical double-layer capacitance (C_dl_) obtained with cyclic voltammetry data at different scan rates (v = 20–100 mV·s^−1^). Stability was determined by CV using 3000 cycles at 100 mV·s^−1^ and by chronopotentiometry at 10 mA·cm^−2^. Overall water splitting tests were carried out in a two-electrode system with the voltage range of 0–2.0 V at a scan rate of 5 mV·s^−1^ in 1.0 M KOH electrolyte.

## 3. Results and Discussion

### 3.1. Characterization of Electrocatalysts

CoMoP nanosheets were produced by a three-step process involving the synthesis of a Co–MOF, its etching and partial cation exchange and a final phosphorization step, as schematically illustrated in Figure 1a. First, a cobalt-based zeolitic imidazolate framework (ZIF-67), consisting of polyhedral-shaped micrometer-size particles, was produced as a self-sacrificial template (Figure 1b). The ZIF-67 was reacted with ammonium molybdate with the double role of etching the structure and partially replacing Co^3+^ cations by Mo^6+^, yielding a porous wrinkled nanosheet-based material that we refer to as Co–Mo MOF (Figure 1c). Finally, the Co–Mo MOF was annealed within a tube furnace containing NaH_2_PO_2_ at 350 °C for 2 h to produce a porous phosphide with a similar wrinkled nanosheet-based morphology that we denoted as CoMoP (Figure 1d and Appendix A). EDS analysis of CoMoP showed a Co–Mo atomic ratio of Co/Mo = 4, and a phosphorus–metal atomic ratio of P/M =2.6 (M = Co + Mo) (Appendix A).

In order to study the effect of ammonium molybdate, we replaced this chemical with an alternative Mo precursor. Using sodium molybdate as Mo source, the morphology of the final product was much more compact, consisting of partially porous cubes (Appendix A). Besides, EDS analysis revealed the molybdenum content of this material to be much lower (Co/Mo = 18.8) than that of CoMoP. We will refer to this material as Mo–CoP. As a reference, a Mo-free CoP was obtained by directly annealing the ZIF-67 in the presence of the phosphorous source, with no etching step (Appendix A). The obtained material also displayed a more compact geometry than that of the CoMoP nanosheets.

Figure 2a,b and Appendix A displays TEM images of CoMoP, further revealing their ultrathin nanosheet structure. HAADF-STEM analysis and EELS chemical composition maps (Figure 2e, Appendix A) displayed a homogenous distribution of C, Co, Mo and P within each CoMoP nanosheet. HRTEM images (Figure 2c) and SAED patterns (Figure 2d) showed CoMoP to present a weak crystallinity, with strong middle/long-range disordered [44,45,46]. In this regard, while the XRD patterns of ZIF-67 and Na_2_MoO_4_-ZIF-67 displayed a good crystallinity (Figure 3a), the Co–Mo MOF already presented a mostly amorphous structure. After phosphorization, CoP maintained a relatively well-organized lattice, and CoMoP displayed a weak crystallographic order, consistently with HRTEM results (Figure 3b).

As expected, the XPS survey spectrum displayed the presence of C, N, O, P, Co and Mo elements on the surface of CoMoP (Figure 3c). The high-resolution P 2p XPS spectrum displayed two doublets, which we associated with P within the metal phosphide lattice (P 2p_3/2_ = 129.7 eV), and a phosphate chemical environment (P 2p_3/2_ = 133.8 eV) (Figure 3d) [41,47]. The Co 2p XPS spectrum displayed six peaks (Figure 3e). The main Co 2p contribution was assigned to Co within the phosphide lattice (Co 2p_3/2_ = 779.3 eV). A second doublet was associated with Co within an oxide, hydroxide or phosphate chemical environment (Co 2p_3/2_ = 781.5 eV). The last two bands were assigned to satellite peaks [28,29,30,48,49]. Finally, the Mo 3d XPS spectrum displayed two doublets assigned to Mo within the metal phosphide lattice (Mo 3d_5/2_ = 228.2 eV) and an oxidized chemical environment (Mo 3d_5/2_ = 233.0 eV) (Figure 3f) [50,51,52].

### 3.2. Oxygen Evolution Reaction

The OER activity of CoMoP was evaluated at room temperature using a three-electrode system in a 1.0 M KOH alkaline solution. As a reference, Mo–CoP, CoP and a commercial RuO_2_ catalyst were also evaluated in the same cell and reaction conditions. The LSV polarization curves displayed the CoMoP to be characterized by an outstanding OER activity, with an overpotential of only 273 mV at a current density of 10 mA cm^−2^ (Figure 4a and Appendix A). This overpotential is well below that of Mo–CoP, CoP and the RuO_2_ electrocatalyst tested here and outperforms that of previously reported CoP-based OER catalysts, as shown in Appendix A. As displayed in Figure 4b, CoMoP was not only characterized by the lowest overpotential at 10 mA cm^−2^ but also provided the lowest Tafel slope, 54.9 mV dec^−1^. This value was significantly below that of Mo–CoP (60.4 mV dec^−1^), CoP (71.5 dec^−1^) and RuO_2_ (86.4 mV dec^−1^), which indicates CoMoP to have associated a faster OER reaction kinetics [53,54].

The electrochemical active areas (ECSA) of CoMoP, Mo–CoP NCs, CoP and RuO_2_ were estimated according to the double-layer capacitance (C_dl_) determined via CV in the non-faradaic region at different scan rates, 20, 40, 60, 80 and 100 mV·s^−1^ (Appendix A and Figure 4c). CoMoP displayed larger C_dl_ (12.6 mF cm^−2^) than Mo–CoP (8.7 mF cm^−2^), CoP (4.9 mF cm^−2^) and RuO_2_ (2.3 mF cm^−2^). This result indicates that CoMoP offers a higher density of accessible electrochemical active sites, which we relate to the proper composition and nanosheet structure of CoMoP.

The charge transport/transfer ability of the electrocatalysts was evaluated by electrochemical impedance spectroscopy (EIS). An equivalent circuit model including a charge transfer resistance (R_ct_) and a solution resistance (R_s_) during the OER process was used to fit the Nyquist plots displayed in Figure 4d [55,56]. CoMoP exhibited the smallest R_ct_ (17.99 Ω), well below that of Mo–CoP (R_ct_ = 27.90 Ω), CoP (R_ct_ = 36.70 Ω) and RuO_2_ (45.47). These results reveal the CoMoP nanosheets to enable a faster charge transfer at the electrode/electrolyte interfaces, thus accelerating the OER electrocatalytic kinetics.

The long-term stability of CoMoP was further analyzed by CV and chronopotentiometry measurements [57,58]. Figure 4e shows how the LSV curve of CoMoP after 3000 CV cycles closely resembles that of the first cycle. The chronoamperometry test displayed CoMoP to have an outstanding long-term catalytic activity with just a 3% current density decay after 100 h of operation at 273 mV (Figure 4f). SEM images of the post-catalysts after OER testing at high current showed the ultrathin CoMoP nanosheets to partially sinter into a highly porous structure with thicker walls (Appendix A). At the same time, the EDX result showed that a loss of P occurred during the OER. These results are consistent with the reorganization of the metal phosphide into a metal (oxy)-hydroxide during the OER reaction [59,60].

### 3.3. Hydrogen Evolution Reaction

The HER performance of CoMoP was evaluated in 1.0 M KOH using a three-electrode system, and it was compared with that of Mo–CoP, CoP and a commercial Pt/C catalyst. As shown in Figure 5a and Appendix A, the CoMoP electrocatalyst displayed a relatively low HER overpotential of 89 mV at the current density of 10 mA cm^−2^, slightly above that of Pt/C (42 mV) and well below that of Mo–CoP (154 mV), CoP (165 mV) and most previously reported phosphide-based HER electrocatalysts (Appendix A). The Tafel slope of CoMoP (69.7 mV dec^−1^) was also much lower than those of Mo–CoP (83.7 mV dec^−1^), CoP (113.4 mV dec^−1^) and close to that of Pt/C (56.1 mV dec^−1^) (Figure 5b), which indicated rapid HER reaction kinetics following the Volmer–Heyrovsky mechanism [61,62]. CoMoP also displayed the smallest semicircular diameter in the Nyquist plot of the EIS data among the phosphide catalysts tested (Figure 5c), showing the lowest charge transfer resistance during catalytic processes. In terms of stability under HER conditions, Figure 5d displays how CoMoP suffered a minor change in the LSV curves after 3000 cycles. Additionally, the CA measurement showed the current density to decrease just 6% after 100 h of continuous operation under HER conditions at an overpotential 89 mV (Figure 5e). The morphology and composition of the catalyst after long-term HER are displayed in Appendix A. In this case, minor changes in structure and a moderate P loss were observed, which points to notable catalyst stability under HER.

### 3.4. Overall Water Splitting

Due to the excellent OER and HER performances demonstrated by CoMoP, a two-electrode configuration electrolyzer with CoMoP both as the positive and negative electrodes was constructed and tested for OWS in 1.0 M KOH solution (Figure 6a). As shown from the polarization curves displayed in Figure 6b, the assembled device just required a cell voltage of 1.56 V to reach a current density of 10 mA cm^−2^, which is significantly below that of a cell containing Pt/C and RuO_2_ electrodes (1.61 V). More importantly, after 40 h of continuous operation at 100 mA cm^−2^, the CoMoP-based cell still maintained an outstanding performance, with just a 14.8% loss at high current density (Figure 6c). Thus, the as-prepared CoMoP can be considered as a highly competitive electrocatalytic catalyst for OWS compared with the previously reported OWS catalysts (Figure 6d, Appendix A). Besides, its outstanding stability demonstrates its potential for large-scale hydrogen production from water splitting.

## 4. Conclusions

In conclusion, ultrathin CoMoP nanosheets were engineered using the Co–MOF ZIF-67 as a self-sacrificial template and ammonium molybdate as a shape-defining agent and Mo source. CoMoP nanosheets exhibited outstanding performance towards HER and OER in alkaline media, which we associate with the proper transport properties and electronic energy levels provided by their composition and their porous nanosheet structure. In particular, CoMoP presented low overpotentials of 89 and 273 mV at a current density of 10 mA cm^−2^ for HER and OER, respectively. Furthermore, CoMoP electrocatalysts also showed excellent long-term stabilities in alkaline electrolytes, with a minor current density decrease after 100 h continuous operation. When used for OWS, a cell voltage of only 1.56 V was needed to reach a current density of 10 mA cm^−2^. This work provides a suitable strategy to synthesize high-performance Co–Mo–P electrocatalysts with abundant exposed active sites and effective avenues for charge and electrolyte transport, and it can be employed to further tune the structure and composition of other 2D nanostructures with optimized performance towards OWS and other electrocatalytic reactions.

## Figures and Tables

**Figure 1 nanomaterials-12-01098-f001:**
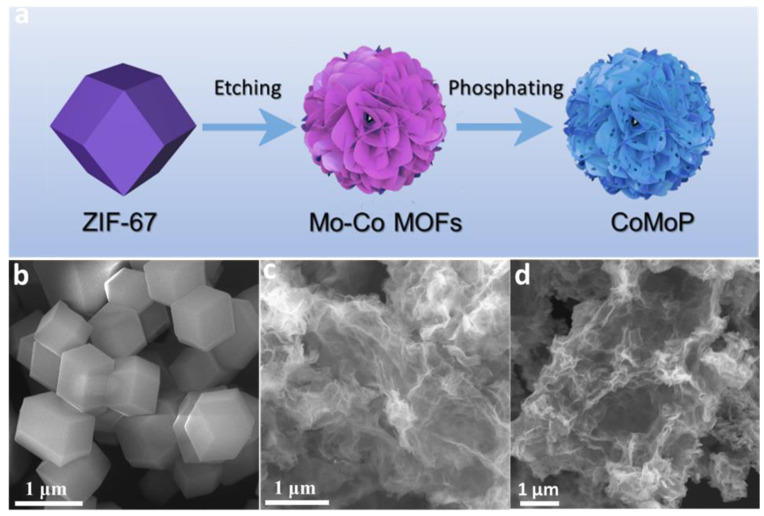
(**a**) Schematic illustration of the CoMoP synthesis process. (**b**–**d**) SEM image of (**b**) ZIF-67 (**c**) Mo–Co MOFs and (**d**) CoMoP.

**Figure 2 nanomaterials-12-01098-f002:**
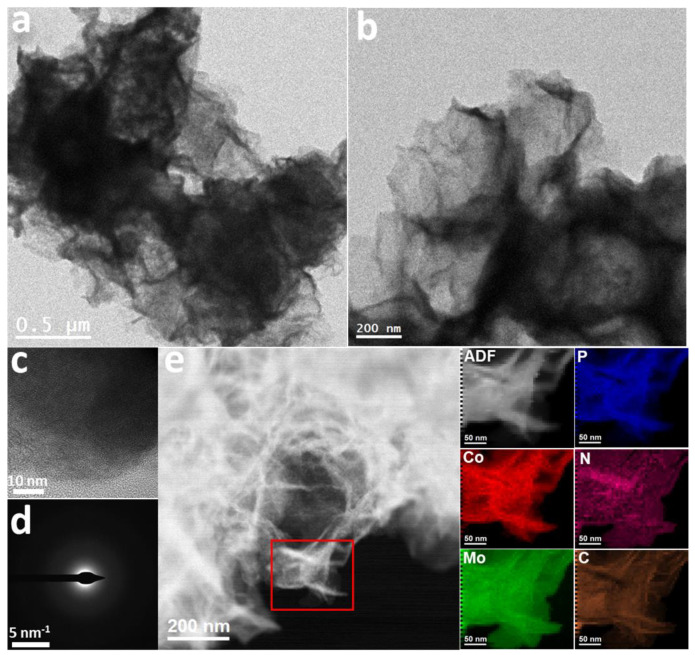
(**a**,**b**) TEM images, (**c**) HRTEM image, (**d**) SAED pattern and (**e**) HAADF STEM image and EELS chemical composition maps obtained from the red squared area of the STEM micrograph. Individual Co L_2,3_ edges at 779 eV (red), Mo M_4,5_ edges at 230 eV (green), P L_2,3_ edges at 132 eV (blue), N K edge at 401 eV (pink) and C K edge at 284 eV (orange).

**Figure 3 nanomaterials-12-01098-f003:**
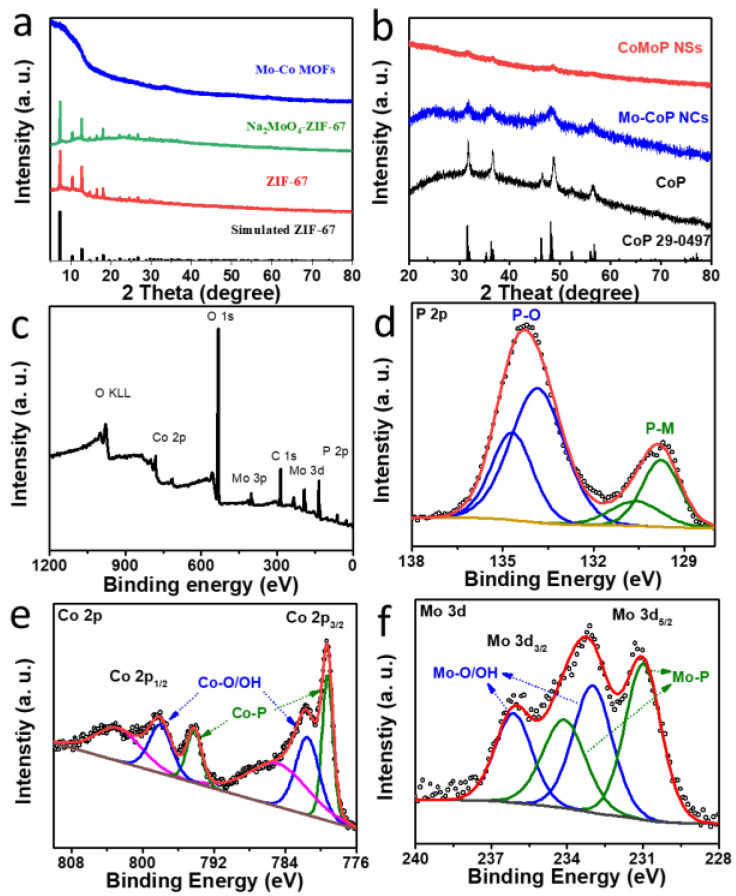
(**a**,**b**) XRD pattern of the MOFs (**a**) and the phosphorized materials, CoMoP, Mo–CoP and CoP. (**c**–**f**) XPS survey and high-resolution P 2p (**d**), Co 2p (**e**) and Mo 3d (**f**) XPS spectra of CoMoP.

**Figure 4 nanomaterials-12-01098-f004:**
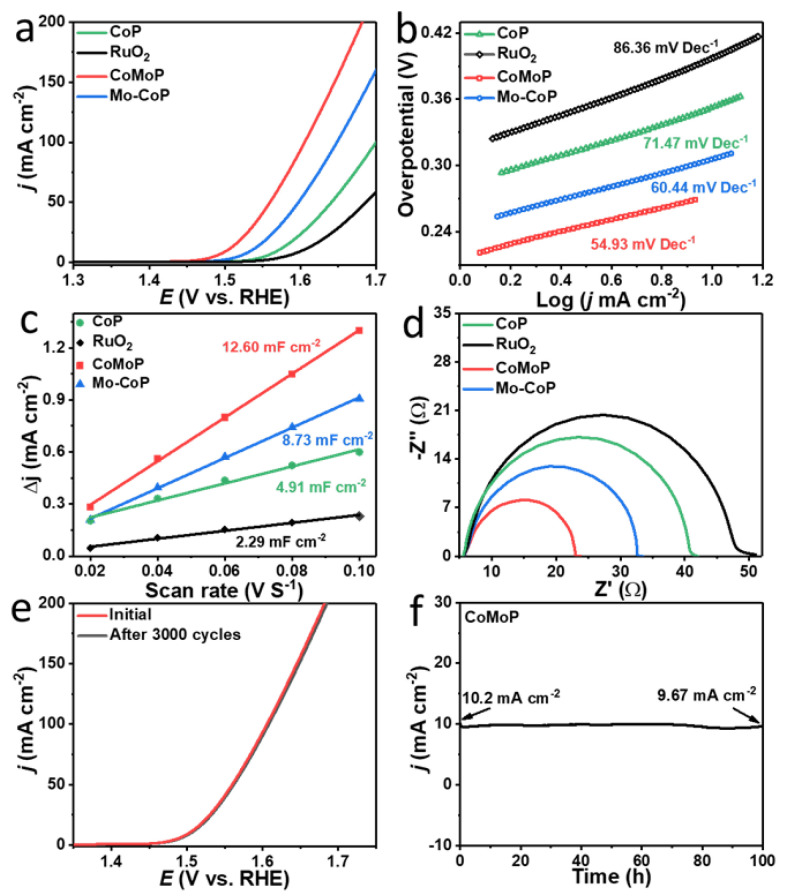
(**a**) OER polarization curves in 1.0 M KOH. (**b**) Corresponding Tafel plots. (**c**) Double-layer capacitances (C_dl_). (**d**) Nyquist plots of the EIS data. (**e**) OER polarization curves before and after 3000 cycles. (**f**) OER chronoamperometric data for CoMoP at an overpotential of 273 mV.

**Figure 5 nanomaterials-12-01098-f005:**
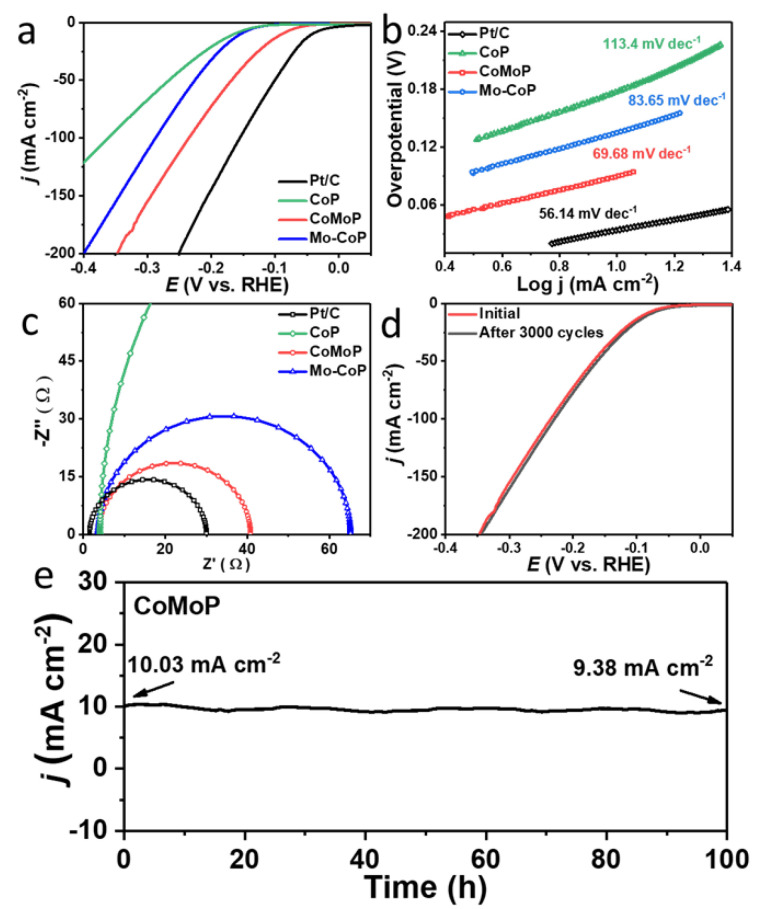
(**a**) HER polarization curves in 1.0 M KOH. (**b**) Corresponding Tafel plots. (**c**) Nyquist plots of the EIS data. (**d**) HER polarization curves before and after 3000 cycles. (**e**) Chronoamperometry data for CoMoP at a 89 mV overpotential.

**Figure 6 nanomaterials-12-01098-f006:**
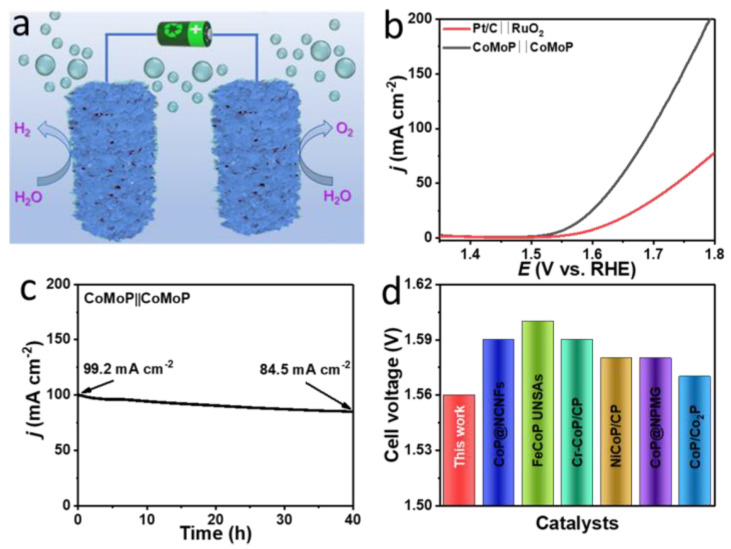
(**a**) Schematic diagram of the OWS in a two-electrode system. (**b**) Polarization curves of OWS cells with the electrode pairs: CoMoP||CoMoP and Pt/C||RuO_2_ in 1.0 M KOH. (**c**) Chronoamperometric curve of CoMoP in the two-electrode system at 1.70 V polarization. (**d**) Comparison of the CoMoP overpotential at 10 mA cm^−2^ with previously reported catalysts in 1.0 M KOH.

## Data Availability

The data are available on reasonable request from the corresponding authors.

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
