# Peer review of "MOF-Derived Ultrathin Cobalt Molybdenum Phosphide Nanosheets for Efficient Electrochemical Overall Water Splitting"

_nanomaterials, 2022, doi:10.3390/nano12071098_

Round 1

Reviewer 1 Report

The article MOF-Derived Ultrathin Cobalt Molybdenum Phospide Nanosheets For Efficient Electrochemical Overall Water Splitting describes the synthesis and use of the ternary materials as electrode for hydrogen and oxygen evolution.

Though the article is of interest to the community to major concerns remain that must be addressed before publication can be considered.

  • The authors use 10 mA/cm2 as metric to report the materials performance and durability. Though it is widely used this metric appears to be outdated and stability evaluation should be performed at relevant currents, ie exceeding 100 mA/cm2
  • A post-catalytic characterization is required. This might be even more evident when tested at higher currents.

Author Response

Reviewer #1: The authors use 10 mA/cm2 as metric to report the materials performance and durability. Though it is widely used this metric appears to be outdated and stability evaluation should be performed at relevant currents, ie exceeding 100 mA/cm2. A post-catalytic characterization is required. This might be even more evident when tested at higher currents.

Response: Thanks very much for your suggestions. Following your advices, we re-tested the stability of the overall water splitting stability at 100 mA cm-2, and the results are shown in Figure 6c.

Figure 6c. Chronoamperometric curve of CoMoP in the two-electrode system at 1.70 V polarization.

The SEM images of post-catalysts after testing at higher currents are shown at Figure S10 for OER. From the result, we can find the ultrathin CoMoP nanosheets to partially sinter into a highly porous structure but with thicker walls. Meanwhile, the EDX result shows a loss of P. These results are consistent with the reorganization of the metal phosphide into a metal (oxy)-hydroxide during the OER reaction. We introduced a discussion in this direction in the revised manuscript, page 7.

Figure S10. a-c) SEM image and d) EDX spectrum of CoMoP after long term OER stability testing.

The morphology and composition of the sample after HER is displayed in Figure S11. In this case, minor changes in morphology were observed and a minor P loss was recorded. This result indicates that the catalyst is very stable during the HER reaction.

Figure S11. a-c) SEM image and d) EDX spectrum of CoMoP after long term HER stability testing.

Reviewer 2 Report

An electrochemical generator designed for the performance of splitting water molecule (or overall water splitting - OWS) is limited by the oxidation of the same molecule, i.e., evolution of molecular oxygen or oxygen evolution reaction (OER). Therefore, the design of materials requires not only catalytic sites, but also the properties of the support material that can modify the electronic properties of these catalytic sites. In this context, the present work presents, in a general way, a material having bifunctional characteristics for the HER and the OER, labeled as a catalyst for OWS. The authors claim that the new catalyst is based on CoP, structured in the form of 2D ultrathin nanosheets and derived from MOF ZIF-67. The authors argue that the porous nanosheet-based structure and the incorporation of Mo within the CoP network allows for rapid water dissociation and efficient and stable HER and OER performance. In sum, such Co-Mo-P electrocatalysts with abundant exposed active sites are effective for charge and electrolyte transfer. There is no doubt that such statements are supported by the results presented in the MS, but from the academic point of view, the rationale for the catalytic activity of the materials in question needs to be further explained. In addition, some inconsistencies need to be reviewed.

  1. Line 184: where is this (Mo 3d5/2 = 228.2 eV) data in Fig 3d (Cf. ref 10.1039/C6EE03768B)? Authors could label the relevant peaks in the XPS figures for the sake of clarity.
  2. As noted above, what is the rational claimed by the authors for indicating that CoMoP is associated with a “faster OER reaction kinetics”?
  3. What is the electrode potential corresponding to the Nyquist plots shown on the MS?

Author Response

  1. Line 184: where is this (Mo 3d5/2 = 228.2 eV) data in Fig 3d (Cf. ref 10.1039/C6EE03768B)? Authors could label the relevant peaks in the XPS figures for the sake of clarity.

Response: Thanks very much for your advices. We label the relevant peaks in the XPS figures. The result is shown below.

Figure 3d. c) XPS survey and high-resolution d) P 2p, e) Co 2p and f) Mo 3d spectra of CoMoP.

  1. As noted above, what is the rational claimed by the authors for indicating that CoMoP is associated with a “faster OER reaction kinetics”?

Response: The faster OER reaction kinetics is concluded from the smaller Tafel slope obtained from CoMoP compared with the reference electrocatalysts and the lower charge transfer resistance obtained from electrochemical impedance spectroscopy (EIS) measurements. 

  1. What is the electrode potential corresponding to the Nyquist plots shown on the MS?

Response: Electrochemical impedance spectroscopy (EIS) was measured in the frequency range of 100 kHz to 0.1 Hz, with an amplitude of 10 mV, and the initial voltage was the overpotential when the current density was 10 mA cm-2. Therefore, the potential was 1.503 V vs. RHE for OER and 0.089 V vs. RHE for HER in the case of CoMoP for example. We included these details data in the experimental section, on page 3

Reviewer 3 Report

Review comments

The manuscript entitled “MOF-Derived Ultrathin Cobalt Molybdenum Phosphide Nanosheets For Efficient Electrochemical Overall Water Splitting” submitted by Wang et. al. In my opinion, this manuscript is interesting to the readers in the field of nanohybrid material synthesis for the energy conversion device application. The manuscript is well-organized and written clearly. Therefore, I recommend it to be published in the Nanomaterials journal.

  1. The author tried to prepare the CoMoP materials with different Mo precursors. However, how did the author predict that the current ratio (Mo: Co) is best for their electrochemical studies?

Author Response

Response: Thank you very much for your comments. We prepared the CoMoP materials with different Mo content through changing the content of ammonium molybdate (50mg, 100mg, 200mg). The OER and HER performance of different Mo content catalysts are shown in the following figure, that is included as Figure S8. According to the result, we got the CoMoP-100 owned the best catalytic performance.

Figure S8. a) OER and b) HER polarization curves of CoMoP with different Mo content in 1.0 M KOH.

Round 2

Reviewer 1 Report

The authors addressed all comments and the article is now publishable